# Bio-Based Polyurethane Composite Foams with Improved Mechanical, Thermal, and Antibacterial Properties

**DOI:** 10.3390/ma13051108

**Published:** 2020-03-02

**Authors:** Sylwia Członka, Anna Strąkowska, Krzysztof Strzelec, Agnė Kairytė, Arūnas Kremensas

**Affiliations:** 1Institute of Polymer and Dye Technology, Faculty of Chemistry, Lodz University of Technology, 90924 Stefanowskiego, Poland; anna.strakowska@p.lodz.pl (A.S.); krzysztof.strzelec@p.lodz.pl (K.S.); 2Institute of Building Materials, Faculty of Civil Engineering, Vilnius Gediminas Technical University, LT-08217 Vilnius, Lithuania; agne.kairyte@vgtu.lt (A.K.); arunas.kremensas@vgtu.lt (A.K.)

**Keywords:** polyurethanes, reinforcement, filler, mechanical properties, antibacterial properties

## Abstract

Among different organic fillers, the chemical composition of *Syzygium aromaticum*, commonly known as cloves, has great potential as a sustainable reinforcement for polymeric materials. In the study, grounded cloves were used as cellulosic filler for a novel polyurethane (PU) composite foams. Soybean oil-based PU composite foams were successfully reinforced with different concentrations (1, 2, and 5 wt%) of clove filler. PU foams were examined by rheological behavior, processing parameters, cellular structure (scanning electron microscopy analysis), mechanical properties (compression test, impact test, three-point bending test), thermal properties (thermogravimetric analysis), viscoelastic behavior (dynamic mechanical analysis) as well as selected application properties (apparent density, dimensional stability, surface hydrophobicity, water absorption, color characteristic). In order to undertake the disc diffusion method, all PU composites were tested against selected bacteria (*Escherichia coli* and *Staphylococcus aureus*). Based on the results, it can be concluded that the addition of 1 and 2 wt% of clove filler leads to PU composite foams with improved compression strength (improvement by ≈18% for sample PU-1), greater flexural strength (increase of ≈11%), and improved impact strength (increase of ≈8%). Moreover, it has been proved that clove filler may be used as a natural anti-aging compound for polymeric materials. Based on the antibacterial results, it has been shown that the addition of clove filler significantly improved the antibacterial properties of PU foams and is suitable for the manufacturing of antimicrobial PU composite foams. Due to these positive and beneficial effects, it can be stated that the use of cloves as a natural filler in PU composite foams can promote a new application path in converting agricultural waste into useful resources for creating a new class of green materials.

## 1. Introduction

Currently, polyurethanes (PUs) are one of the most important groups of polymeric materials [1]. Among different types of PU materials, polyurethane foams correspond to 67% of total PU consumption. Among commercially available insulating materials, such as mineral wool or expanded polystyrene, rigid PU foams exhibit better insulation properties. The thermal conductivity coefficient (λ) of PU foams varies between 0.018 and 0.025 W·m^−1^·K^−1^. Due to this, rigid PU foams are commonly used in varied applications, such as construction or building, industrial insulation, or household appliances [2]. 

The main problem associated with the PU industry is the dependence of isocyanates and polyols on petrochemical products. Legislative requirements connected with environmental protection lead the chemical industry towards the development of new green composite materials [3,4,5]. One of the solutions that meet these regulations is the introduction of polyols derived from natural sources, such as vegetable oils [6,7,8,9]. Several studies have reported bio-polyols based on different vegetable oils, such as castor oil [10], soybean oil [11,12], palm oil [13,14], rapeseed oil [15,16], tung oil [17], sunflower oil [18] or canola oil [19] to develop a new kind of environmentally friendly bio-based PU foam [20,21].

The main disadvantage of PU foams based on bio-polyols is relatively low mechanical strength [22]. It has been stated that the addition of different kinds of organic and inorganic fillers may enhance the mechanical properties of PU composite foams. In the literature, many different inorganic fillers, such as nanoclay [23], expandable graphite [24], silica [25], talc [26], or polyhedral oligosilsesquioxanes (POSS) [27,28] has been already used in the production of PU foams. Besides numerous favorable properties of such prepared PU composites, the application of the fillers obtained from natural sources has attracted much more attention. Incorporating bio-fillers into PU foams improves the environmentally friendly character and the mechanical properties of PU foams [29,30,31,32]. Until now, many different natural fillers have been investigated. For example, Zhou et al. [33] produced rigid PU foams modified with cellulose nanocrystals. PU composite foams with enhanced mechanical properties, reduced water uptake, and greater dimensional stability were obtained. Paberza et al. [34] synthesized PU composite foams enhanced with wheat straw lignin at different concentrations (0–6.3 wt%). Improved thermal insulation properties were obtained for PU composites containing 3 wt% of the filler. Zieleniewska et al. [35] synthesized PU composite foams enhanced with eggshell. The incorporation of eggshells into the PU matrix improved the mechanical properties, reduced water uptake, and increased dimensional stability in selected aqueous media. The influence of potato protein, buffing dust, and keratin feathers in a certain amount on the morphology as well as physical and mechanical properties of the obtained porous materials were tested in our previous study [36,37,38]. It has been shown that a small addition of the fillers (such as 1–2 wt%) improved the compressive strength of material by about 20%, due to the strengthening effect of the fillers. 

Among different types of fillers used as a reinforcing filler for PU foams, up to date there are no publications that were carried out to examine the enhancement of mechanical, thermal, and antibacterial properties of PU composite foams modified with the addition of *Syzygium aromaticum*, commonly known as cloves. Nowadays, cloves are cultured in different parts of the world including Indonesia or Brazil. Cloves are one of the richest sources of the phenolic extract [39]. It consists mainly of eugenol (50%–90%), eugenol acetate, thymol, and tanene. Due to this, cloves possess outstanding properties including antioxidant, antimicrobial, and antifungal activity. Clove extract is widely used as an additive in polymeric materials, for example, food packaging materials. 

To the best of our knowledge, cloves as a filler in the PU foams have not been used to date. Keeping in view the outstanding properties of cloves, it seems logical to use cloves as a reinforcing filler for the new bio-based PU composite foams. The use of cloves as a reinforcing filler may improve the mechanical and physical properties of PU composite foams. It is expected that the incorporation of the clove filler may improve the anti-aging and antibacterial properties of the obtained materials. Due to these positive and beneficial effects, it can be stated that the use of cloves as a natural filler in PU composite foams will promote a new application path in converting cloves into valuable resources for producing a new class of green materials.

Therefore, this study examines the influence of clove filler as a new reinforcing filler for PU composite foams that could be suitable for production according to the requirements of European standards. The current study focuses on the application of clove filler as a natural bio-filler to obtain PU composite foams with enhanced mechanical, thermal, as well as antibacterial and anti-aging properties. PU composite foams enhanced with different concentrations (1, 2, and 5 wt%) of clove filler were examined by means of antibacterial test against selected bacteria (Disc Diffusion Method). The influence of the filler addition on mechanical properties (e.g., compression test, bending test, impact test, dynamic-mechanical behavior), thermal properties (e.g., thermogravimetric analysis), application properties (apparent density, water absorption, dimensional stability), and morphology (porosity, cell size distribution, cell size diameter) of PU composite foams was examined. Based on the results, it may be concluded that the incorporation of clove filler in the concentration of 1–5 wt% influences the rheological behavior of polyol premixes, leading to improvement or deterioration of the aforementioned properties of clove-based PU composite foams.

## 2. Experimental 

### 2.1. Materials and Manufacturing

IZOPIANOL 30/10/C was purchased from Purinova Sp. z o.o. It is a commercial mixture of polyester polyol (functionality of 2, hydroxyl number of 230–250 mg KOH/g), flame retardant (Tris(2-chloro-1-methylethyl)phosphate), catalyst, and chain extender (1,2-propanediol). PUROCYN B was purchased from Purinova Sp. z o. o. It is a polymeric diphenylmethane 4, 4’-diisocyanate (pMDI) (31 wt% of isocyanate groups). 

This formulation was selected because its industrial relevance for the production of PU insulation ensures that the raw materials are available from a range of manufacturers at a competitive cost. Both commercial components were combined in a ratio of 100:160 (ratio of OH:NCO groups), in pursuance of the information provided by the producer of the component. 

ERGOPLAST^®^ES was supplied by Boryszew S.A. It is a commercial soybean oil-derived polyol (hydroxyl number of 156 mg KOH/g). Cloves were purchased from Green Essence Sp. z o.o. The formulations of the PU foams are shown in Table 1. 

In the first step, a mixture of PUROCYN B and grounded clove filler in an amount of 1, 2, or 5 wt% was stirred for 60 s at a speed rate of 4500 RPM. Then, petroleum-derived polyol (IZOPIANOL 30/10/C) and soybean oil-derived polyol (ERGOPLAST) were added to the mixture in a selected amount. The mixture was homogenized at 4500 RPM for approximately 60 s and poured into the open mold. PU composites were conditioned for 24 h at room temperature. A schematic figure of the synthesis of PU foams is presented in Figure 1. 

### 2.2. Characterization Techniques

The average size of the clove filler was determined by the dynamic light scattering (DLS) method using a Zetasizer NanoS90 instrument (Malvern Instruments Ltd, Malvern, UK). The dispersion of the clove’s particles in a polyol dispersion was prepared (0.04 g·l^−1^). The measurement was evaluated at 5 min intervals.

The viscosity of the polyol premixes modified with clove filler was determined according to ISO (International Organization for Standarization) 2555 using a Viscometer DVII+ (Brookfield, Dresden, Germany). The measurement was performed in the function of a shear rate (0.5 to 100 s^−1^) at ambient temperature.

The chemical structure of cloves was determined by Fourier-transform infrared spectroscopy (FTIR) using Nicolet iS50 FTIR Spectrometer with a DGTS/KBr detector (Thermo Fisher Scientific, Waltham, MA, USA). The measurement was performed for the wavelength range from 3500 to 400 cm^−1^ with a maximum resolution of 4 cm^−1^. FTIR was performed with a DGTS/KBr detector. The presented FTIR spectrum is an average of 64 individual scans. 

The morphology of PU foams was determined by JEOL JSM-5500 LV scanning electron microscopy (JEOL Ltd., Tokio, Japan). The samples were scanned in the free-rise direction at the accelerating voltage of 10 kV and magnification of 50 µm. Statistical analysis of the size distribution of the PU foam cells was calculated on the basis of SEM images using ImageJ software (Java 1.8.0_112, Media Cybernetics Inc., Rockville, MD, USA).

The apparent density of PU foams was determined as the ratio of sample weight to its volume in accordance with ISO 845 standard.

The content of closed cells of PU composite foams was determined in accordance with ISO 4590 standard.

Compressive strength (σ_10%_) of PU composite foams was measured according to ISO 844 standard using Zwick Z100 Testing Machine (Zwick/Roell Group, Ulm, Germany) at a constant speed of 2 mm·min^−1^ and load cell of 2 kN to 10% of deformation. PU composite foams were tested perpendicular and parallel to the foam rise direction.

Flexural strength (ε_f_) of PU composite foams was measured according to ISO 178 standard using Zwick Z100 Testing Machine (Zwick/Roell Group, Germany) at a constant speed of 2 mm·min^−1^. 

Surface hydrophobicity of PU composite foams was measured using contact angle goniometer OEC-15EC (DataPhysics Instruments GmbH, Filderstadt, Germany) with software module SCA 20. A total of 1 μL of water was dropped on the surface of PU samples using a micrometer syringe with a steel needle. The average of 10 measurements was evaluated. The surface energy of PU foam composites was calculated on the basis of measured contact angles. 

The thermal properties of PU foam composites were determined by thermogravimetric analysis (TGA) using STA 449 F1 Jupiter Analyzer (Netzsch Group, Selb, Germany). The measurement was performed for samples of 10 mg. Samples were heated in an argon atmosphere up to 600 °C. The initial decomposition temperatures, such as T_10%_, T_50%_, and T_80%_ of mass loss were determined. 

Water absorption (WA) of PU foam composites was performed according to ISO 2896. Samples were weighed (m_0_) and immersed in distilled water for 24 h (water depth of 1 cm). After this time, PU foams were removed from the water and the excess of water was absorbed by filter paper. PU samples were weighed again (m) and the water absorption was calculated according to Equation (1).
(1)$$\mathrm{WA}=m-m_{0}/m_{0}$$

Color measurement was evaluated to determine the color change of PU composite foams before and after aging. The color change was performed using a CM-3600d spectrophotometer (Konica Minolta Sensing, Japan) in the wavelength range of 360–740 nm. The value of total color change (∆E*) was calculated in accordance with Equation (2), where (L*) is brightness, (a*) is a red-green component, and (b*) is a blue-yellow component.
(2)$$\Delta E=\sqrt{{\left(a^{*}\right)}^{2}+{\left(b^{*}\right)}^{2}+{\left(L^{*}\right)}^{2}}$$

Aging in a climate chamber was performed using a UV 2000 apparatus (Atlas Material Testing Solution, USA). The destructive parameters, such as humidity, temperature, and UV radiation were examined. PU samples were conditioned in a climate chamber with the following parameters: temperature of 70 °C, the relative humidity of 70%, and radiation intensity of 0.7 W m^−2^. Samples were conditioned in a climate chamber for 7 days. After this time, an effect of combined, destructive parameters (temperature, humidity, and UV radiation) on selected properties of PU foam composites (e.g. color change, cellular structure) was evaluated. 

Antibacterial properties of PU foams against *Escherichia coli* (*G−*) and *Staphylococcus aureus* (*G+*) were examined according to the National Committee for Clinical Laboratory Standards [40] by Disc Diffusion method. Selected bacteria were mixed with the growth medium (agar) at a temperature of 40 °C, poured on Petri dishes and solidified. After 30 min PU sample with a diameter of 8 mm was placed on the solidified agar. Such prepared foams were incubated in a hermetic thermostat at 37 °C for 24 h. After this time, optical images of the bacterial growth zone around the PU were made and the results were measured.

## 3. Results and Discussion

### 3.1. Characterization of Clove Filler

The analysis of UV-Vis (Ultraviolet–visible) spectra (Figure 2) confirmed the presence of phenolic compounds of clove filler. The maximum absorbance of eugenol occurs in the range of 270–300 nm. The low-intensity peak at 320–330 nm may be ascribed to flavones and their glycosides. The peak at 490 nm indicates the presence of colored compounds, such as chlorophyll or carotenoids. It is also confirmed that the chlorophylls a and b exhibit the absorbance peak between 400 and 500 nm as well as in the range of 600–700 nm [41,42].

The functional groups of clove filler were confirmed by FTIR spectroscopy (Figure 3). The characteristic bands occur at 3448 cm^−1^ (–OH vibration) and 2920 cm^−1^ (–CH_2_ vibration) and they correspond to the presence of cellulose, hemicellulose, and lignin. The intense band at 1630 cm^−1^ may be attributed to the mixed vibration of (C=O) and (C=C) [43]. The shoulder peak localized at 1602 cm^−1^ indicates the presence of symmetric aromatic ring (C=C) [43]. Bands at 1610-1600 cm^−1^ (C=C vinyl stretching mode), 1480–1450 cm^−1^, and 1515 cm^−1^ (both bands related to C=C aromatic stretching modes) are associated with the chemical structure of eugenol [44]. Other characteristic bands of the phenol group correspond to 1370–1310 cm^−1^ (C–OH deformation vibrations) as well as 1170–1110 cm^−1^ (C–OH stretching vibration) [39].

The incorporation of clove filler influences the rheological behavior of polyol mixtures. The dispersions of 1, 2, and 5 wt% of clove filler in polyol mixtures are presented in Figure 4. The optical images indicate that clove particles are not dissolved completely in the polyol system. At high concentration of the filler, such as 5 wt% (Figure 4c), the clove particles are bound to each other and they tend to agglomerate due to the high energy and surface area of the particles. 

The agglomeration of the filler particles was also confirmed by the results of the particle size distribution (Figure 5). The particle size of the filler was measured in a polyol dispersion (1 g/100 g of polyol). The measurements were done with the time interval between every measurement. The particle size distribution of the sample measured after 5, 10, 15, 20, and 25 min of ultrasonic mixing are given in Figure 5. The results presented in Figure 5 indicates that filler particles tend to agglomerate. After 25 min, the size of particles increases from 1.6 to 3 µm. This indicates that during the measurement, the particles tend to agglomerate and the presence of bigger agglomerates in a polyol system may be observed.

Table 2 presents the results of dynamic viscosity depending on the concentration of clove filler in polyol mixtures. Increasing the concentration of clove filler particles from 1 to 5 wt% increases the viscosity of polyol mixtures. For example, compared to unmodified polyol, the dynamic viscosity of mixtures increases by ≈185% at 5 wt% of clove filler. The viscosity of all modified systems decreases in the function of share rates. This dependence is well-known for non-Newtonian liquids with a pseudo-plastic nature. An analogous relationship has also been observed for other organic fillers. It was stated that an increased viscosity of modified systems may be related to high interaction between polyether polyol and filler particles through hydrogen bonding and van der Wall’s interaction [45,46].

### 3.2. Foaming Behavior of PU Mixtures Containing Clove Filler

The foaming process of PU foam composites was monitored by measuring appropriate processing times, such as cream time, expansion time, and tack-free time. 

As presented in Table 3 an incorporation of clove filler leads to prolonged cream and extension time. Compared to PU-0, cream time increases from 43 to 60 s, while expansion time increases from 214 to 238 s for sample PU-5. It has been well described in previous works that well-dispersed filler particles may act as an additional nucleation center leading to the formation of a greater number of bubble cells during the nucleation process. On the other hand, as discussed previously, the incorporation of clove fillers influences the viscosity of the initial system, which affects the foaming process. The viscosity of the modified systems is increased, and the further growth of formed cells is limited. The foaming process may be also disturbed by the addition of clove extract with reactive groups, which react with isocyanate, creating an imbalance in the ratio of NCO to OH groups. This dependence is also confirmed by a decrease in the maximum temperature (T_max_) measured during the foaming process. It can be concluded that the incorporation of clove filler leads to lower reactivity of the modified systems. On the other hand, lower values of T_max_ may be connected with the fact that clove filler may absorb the heat produced during the reaction between NCO and OH [47]. Due to this, the cream and extension time of the modified systems is increased. The incorporation of clove filler in each amount leads to reduced tack-free time. This indicates that clove particles may act as a curing accelerator during the foaming process.

### 3.3. Cellular Structure of PU Composite Foams

Cellular morphology influences the mechanical and thermal properties of porous materials [48,49]. A proper correlation between filler dispersion and the viscosity of the PU premixes is the main factor that determines the morphology of the obtained materials. The effect of different amounts of clove filler on the morphology of PU composite foams was examined (Figure 6). The incorporation of solid particles seems to affect the morphology of PU composite foams. Among the modified foams, the morphology of PU-1 is more homogenous with a well-preserved closed-cell structure. With increasing filler content, the morphology becomes less uniform and the higher number of damaged cells is observed in the structure of PU foams. The morphology of PU-5 is characterized by the presence of small cavities that break up to generate a formation of large holes. The results presented in Table 3 indicate that depending on clove filler concentration, the porosity of PU foam composites decreases from 91% to 87% for samples PU-1 and PU-5, respectively. This phenomenon was well-described in previous studies and it is attributed to the poor interphase interaction between PU matrix and filler surface, which promotes cell collapsing of crowded cells leading to the formation of open cells [50,51,52]. The formation of open cells is enhanced by possible interaction between the PU matrix and clove filler, which disturbed the formation of stable cellular structure. It is also visible (Figure 6) that the filler particles are not well built in the cell struts, however, some particles are also visible in voids. This indicates that the incorporation of the high amount of clove filler results in the formation of some aggregates, which, in turn, promotes the formation of broken cells. 

It may be also observed (Figure 7) that the incorporation of clove filler leads to the structure with smaller cells, however, the incorporation of the high amount of clove filler, such as 5 wt%, leads to the higher frequency and wider distribution of cell sizes, as compared to PU-0. Samples PU-1 and PU-2 possess pores with an average diameter in the range of 200–300 µm, while for sample PU-5, two populations of pore diameter are observed—a large one located at ≈800 µm, and a smaller one with diameter ≈200 µm.

This indicates that the cell diameter transitions from a unimodal distribution to a bimodal distribution due to the higher content of open pores present in the structure. As discussed previously the incorporation of the high content of clove filler, such as 5 wt%, results in poor interphase interaction between PU matrix and filler surface, which promotes cell collapsing of crowded cells. Moreover, previous studies have shown that filler particles can act as gas nucleation sites during the foaming process and assist the formation of nucleation centers for the gaseous phase [26,53], thus affecting local rheology surrounding the growing bubbles [54]. The addition of powder filler can change the nucleation mode from homogenous to heterogeneous and reduce the nucleation energy, which in turn promotes the formation of large numbers of small cells [26], increasing the tendency of cell coalescence and leading to higher inhomogeneous cell size distribution. By controlling the type of nucleation, a vast range of cell sizes can be covered. Previous studies have shown that the morphology depends on the concentration and molecular mass of both soft and segments [55,56]. Thus, the incorporation of clove filler may change the crosslinking density of PU materials as a result of the incorporation of additional groups of clove fillers, which can react with isocyanate groups. Due to this, the molecular mass of both soft and hard segments can change leading to different morphology structures of the obtained materials (Figure 8). Similar dependence has been observed in the case of different kinds of PU materials [55,56]. Whereas the morphology of PU materials is an important factor that affects many materials properties, it is mandatory to study more in-depth the nucleation and growth process of PU systems to aid understanding and optimization of their materials behaviors. 

### 3.4. Apparent Density of PU Composite Foams

An insignificant tendency to increase the apparent density is observed for PU composite foams. As presented in Table 3 the value of apparent density of PU-0 and PU composite foams depends not only on the concentration of the clove filler but it is also affected by the position of the PU foams and it changes among different parts (upper/middle/bottom). It has been stated in previous studies that the apparent density is affected by the density gradient of the filler [57]. Due to this, the lowest value of the apparent density is observed for the bottom part of PU foams, which is characterized by an open-cell structure with visible voids and broken cells (Figure 9). The middle part of PU foams is characterized by the highest values of apparent density as a result of a more homogenous structure with a great number of closed-cells. Apparent density decreases for the upper part of PU foams due to more intensive expansion and the presence of cells with a larger diameter.

Basically, the incorporation of clove filler leads to an increase in apparent density. For example, the value of apparent density measured for the middle part of PU foams, increases from 41 to 44 kg m^-3^ for samples PU-1 and PU-5, respectively. Beside the fact that PU foams modified with higher content of clove filler (such as 2 and 5 wt%) are characterized by a more porous structure, the incorporation of solid particles with a specified density (density of powdered clove ≈1.4 g·cm^−3^) is a prominent factor that determines the density of the obtained materials. Similar dependence has been observed in previous studies [58,59,60]. 

### 3.5. Compression Strength, Flexural Strength, and Impact Strength

Apparent density strongly affects the mechanical properties of porous materials [46,61]. Compression strength (σ_10%_) of PU composite foams was measured in a parallel and perpendicular direction in relation to the foam growth (Table 4). Compared to PU-0, the incorporation of 1 and 2 wt% of clove filler leads to significant improvement of σ_10 %_. The value of σ_10%_ (measured parallel) increases by ≈18% and ≈13% for samples PU-1 and PU-2, respectively. No further improvement of σ_10%_ is observed for sample PU-5. Comparing to PU-0 the value of σ_10%_ decreases by ≈16%. An analog tendency is observed in the case of specific compression strength, which means that density is not the main factor that influences the mechanical properties of PU foams. Another factor may be found in characteristic features of PU foam structure.

The incorporation of clove filler influences the values of flexural strength (σ_f_) and impact strength of PU composite foams. Compared to PU-0 the aforementioned properties are improved, however, with increasing filler content the mechanical properties tend to decrease. Comparing to PU-0, σ_f_ increases by ≈11%, ≈9%, and ≈2% for samples PU-1, PU-2, and PU-5, respectively, while the impact strength increases by ≈18%, ≈11%, and ≈8%. Solid particles of clove filler act as reinforcing centers and generate localized stresses under the action of a loading force. Energy dissipation takes place when a growing crack encounters a filler particle distributed in a reinforced polymer matrix. The number of filler particles increases as the filler concentration increases. During bending, an excess of solid particles acts as stress concentration centers that generate damage and cracking of the sample leading to the deterioration of the mechanical properties. The elongation of PU composite foams seems to be inversely proportional to the strength. With increasing concentration of clove filler, PU composite foam exhibits a prolonged range of elongation that may be attributed to the greater flexibility of polymer matrix due to the higher content of open cells. The analog trend has been also found in previous works [62,63].

### 3.6. Dynamic Mechanical Analysis of PU Composite Foams

The dynamic mechanical analysis of PU composite foams as a function of temperature (0–600 °C) is presented in Figure 10. Examination of tanδ values provides information about the damping properties of PU composite foams. The maximum value of tanδ corresponds to the glass transition temperature (T_g_). Based on the results of T_g_ (Table 4) it can be concluded that the incorporation of 1 and 2 wt% of clove filler does not affect the value of T_g_. The lower value of T_g_ is observed for sample PU-5, indicating that that the addition of 5 wt% of clove filler leads to decreased cross-linking density and increased mobility of the chain segments. Previous studies have shown that the value of T_g_ is dependent on the aromaticity and cross-linking density of PU materials [64]. Based on this, it can be concluded that the incorporation of clove filler in high concentration, such as 5 wt%, decreases the cross-linking density and increases an empty fraction in PU structure as a result of limited interaction between PU chains. Due to this, the mobility of PU chains is increased and the value of T_g_ is reduced. On the other hand, the incorporation of clove filler may decrease the reactivity of the PU system due to the high viscosity of the PU mixture and disturb the foaming process. The reactive groups (i.e., hydroxyl groups) of clove filler may react with isocyanate groups, leaving fewer isocyanate groups available for the reaction with polyol. Due to this, the number of thermally stable polyurethane bonds is reduced. An analog trend has been also reported in previous studies [17,65]. 

The incorporation of 1 and 2 wt% of clove filler improves the thermo-mechanical properties of PU composite foams. As presented in Figure 9b, samples PU-1 and PU-2 exhibit higher value of storage modulus (E’) as compared to PU-0. With increasing clove filler content up to 5 wt% the value of E’ decreases. The thermomechanical behavior of PU foams is a complex effect that involves a reinforcing effect of the filler and the structure of PU foams. In the case of PU-1 and PU-2, an improved thermo-mechanical behavior results from the rigid structure of clove filler and well-developed closed-cell structure. Further increase of clove filler content results in an open-cell structure with a higher number of broken cells. It may be concluded that the incorporation of rigid particles of clove filler is not sufficient to compensate for negative changes in the morphology of PU foams. This results in deterioration of the thermomechanical stability of PU composite foams.

### 3.7. Thermogravimetric Analysis of PU Composite Foams

Thermogravimetric analysis (TGA) was used to evaluate the effect of clove filler on the thermal properties of PU composite foams. The results of the TGA analysis are presented in Table 5 and Figure 11. Characteristic temperatures were defined as T_10%_ (temperature corresponding to 10% of weight loss), T_50%_ (temperature corresponding to 50% of weight loss), T_80%_ (temperature corresponding to 80% of weight loss), and T_max_ (temperature corresponding to the highest weight loss).

Compared to PU-0, the composite foams modified with clove filler are less thermally stable. It is noticed on the thermograms of PU foams that a few percent of weight loss occurs around ≈50 °C which may be attributed to the migration of blowing agents in the PU matrix. The degradation of PU-0 starts at 254 °C, while the degradation of PU composite foams starts in the range of 226–238 °C, depending on the concentration of clove filler. The highest loss of weight for sample PU-0 occurs in the range of 200–600 °C (the rate of weight loss ≈5.5 wt% min^−1^), while for PU composite foams it occurs in the range of 200–550 °C (the rate of weight loss in the range of 4.5–5.4 wt% min^−1^) depending on the concentration of the clove filler. Reduced range of the temperature attributed to the highest weight loss is mostly related to an organic character of clove filler and thermal degradation of the organic compounds contained in the clove filler, such as protein, flavonoids, and oils, which decompose completely during heating. 

A slight decrease in thermal properties can be attributed to the uneven distribution of the filler in a polymer matrix which affects the crosslinking density [44]. Moreover, the results of TGA are also in agreement with the morphology of PU composites presented in Figure 6. Higher content of open voids, which are presented in the structure of modified foams, accelerates the degradation process leading to the slight deterioration of thermal properties of analyzed materials. On the other hand, the results of TGA indicates that the incorporation of clove filler decreases the weight loss of PU composites during the initial step of degradation. Due to the barrier effect of incorporated filler particles, the oxygen and heat fluxes in the direction of the polymer surface are limited and the weight loss is reduced. 

The difference between pure PU foam and PU composite foams is also confirmed by DTG analysis which is defined as the first derivative of TGA. DTG analysis provides information about the decomposition rate of PU foams during the heating process. All samples show a similar three-steps degradation path with maximum degradation intensity in the range of 320–330 °C, which is attributed to the degradation of hard segments of PU. In the case of foam composites an additional, slight overlapped influx occurs at ≈325 °C. This may be connected with faster degradation of the urethane bond, which is formed between urethane groups and hydroxyl groups of the clove filler surface. The introduction of organic filler, which may contain small amounts of residual water, results in the formation of more urea bonds, increasing the activation energy of thermal decomposition as compared to unmodified foam. 

### 3.8. Microbial Properties of PU Composite Foams

The microbiological test was performed to evaluate the influence of clove filler on the antibacterial properties of PU composite foams. The results of the microbiological test are presented in Table 6. A significant elimination of selected bacteria (*Escherichia coli* and *Staphylococcus aureus*) is observed after 24 h of exposure. The best antibacterial results are obtained for a sample containing 5 *wt%* of clove filler. The inhibition of selected bacteria (*Escherichia coli* and *Staphylococcus aureus*) by application of 5 wt% of clove filler is about 76% and 79%, respectively. Antibacterial activity of the samples containing clove filler may be attributed to the phenolic content in the PU foams, which can induce a strong fungicidal and microbicidal effect, protecting samples against bacteria and fungi. 

### 3.9. Color Analysis of PU Composite Foams

The color change is the main parameter, which is a visual indicator of changes occurring in the polymer matrix due to the degradation process [66]. PU composite foams were evaluated optically to determine the color change after the UV aging. The results of the color analysis are presented in Table 7. The addition of clove filler influences the color of the obtained PU composite foams. A significant difference in total color change may be seen after the addition of the clove filler. Compared to PU-0, with increasing concentration of clove filler, PU composite foams are characterized by a reduced value of L* indicating the more intense color of PU composite foams. Increased values of a* and b* pointed out that comparing to PU-0, modified samples possess more intense yellow and red shades.

The results presented in Table 8 indicate that clove filler may be successfully used as an organic indicator of color change after the aging process. The color changes were evaluated in terms of the color parameters. According to the literature, the photodegradation process of organic compounds is mainly associated with the oxidation of chromophore groups and, consequently, with depolymerization of polymer chains [67]. After the UV aging, organic fillers normally tend to lose red (< a*) and yellow (< b*) shades as a result of oxidation of organic compounds present in the PU chain. Due to this, increased values of a* and b* may be ascribed to the oxidation of natural compounds present in the PU chain. This applies mainly to amines that are formed in the reaction between water and isocyanate groups after UV exposure, which in turn transforms into quinones.

The most visible change of total color changes (ΔE*) after the UV aging process is observed for PU-0 (Table 8). With the increasing content of clove filler, the difference in ΔE* is reduced. It can be concluded that the addition of clove filler, which is a natural antioxidant compound, may protect the PU composite foams from UV radiation and high temperature. The incorporation of clove filler improves the stabilization of the PU composite foams and may be used as an anti-aging compound in the production of PU foams. The high resistance of cloves against high temperature and UV radiation should be justified by the chemical composition of cloves, including various extracts, such as eugenol, ether, or methanolic extracts which possess strong antioxidant activities and prevent a natural discoloration of PU composite foams.

An organoleptic examination confirmed that after the UV radiation exposure, the surface color of PU foams changed to orange even before the addition of clove filler (Figure 12 and Figure 13). Discoloration of PU foams is mostly related to the oxidation process of aromatic amines which form during the reaction of isocyanate and water. Oxidation of aromatic amines leads to the formation of chromophores which contribute to discoloration of PU foams. Moreover, the discoloration of PU materials is affected by the presence of aliphatic polyesters and aliphatic polyethers, which are easily susceptible to the oxidation process. In all cases, the surface of PU foams becomes less brittle after UV radiation exposure than the surface of PU-0. The surface of PU-0 subjected to one week of degradation is more destroyed and a higher number of cracks and holes are observed, as compared to PU foams with the addition of clove filler (Figure 14). This indicates that the incorporation of clove filler may improve the strength of the bonds responsible for reducing the brittleness and improve the aging properties of PU composites. 

### 3.10. Contact Angle, Water Absorption, and Dimensional Stability

The comparison of the results clearly indicates that after the degradation process samples PU-1 and PU-2 are characterized by lower hydrophobicity, which is confirmed by a decrease in the contact angle (Table 9). An opposite effect is observed for sample PU-5, which contains the highest concentration of clove filler in the amount of 5 wt%. The possible explanation may be found in the fact that during the photodegradation process, the reactive groups present in the structure of clove filler are able to induce a lower adhesion force, changing the surface energy of the PU composite foams and hindering the spread of water droplets. It seems that the particles of the clove may act as photocatalytic agents and protect the foams from the photodegradation process, however, this statement is not yet fully understood and further research on this topic should be undertaken. 

It is well known from the literature, that water uptake depends both on the morphology of PU foams (open/closed-structure) and the hydrophobic character of the incorporated additives [35]. As presented in Table 9, the addition of clove filler influences the water uptake of PU composite foams. With increasing the concentration of clove filler, the water uptake increases, due to more open-cell structure of PU foams. As presented in SEM images (see Figure 6) the incorporation of the high content of clove filler, such as 5 wt%, affects the opening of the foam cells. Thus, the broken cells of modified PU composite foams are able to accommodate more water as compared to PU foams with a well-developed closed-cell structure. Based on this, it is clear that the morphology of porous materials is the main factor that determines the water uptake of the PU composite foams. Greater water absorption of PU composites may be also ascribed to the hydrophilic nature of clove filler [39]. In the general trend, it can be seen that with increasing concentration of clove filler the contact angle (measured between sample and water) slightly decreases. This may be related to the hydrophilic nature of the clove filler. Among different compounds, cloves contain a high number of polyphenols with –OH groups. Due to this, the character of PU composite foams containing clove particles becomes more hydrophilic as compared to the unmodified PU-0. A similar trend was also described in previous works [36]. In this case, water uptake of analyzed PU foams is insignificantly higher, however, the results are still satisfactory from an application point of view.

The linear changes in dimensions, volume, and mass of PU foams after conditioning at −20 and +70 °C for up to 14 d were examined. The results of aging measurements are presented in Table 9. Compared to PU-0 the addition of clove filler results in insignificant changes in linear dimensions (Δl), volume (ΔV), and mass (Δm). After the special treatments, the measurements of changes in Δl, ΔV, and Δm are random and no correlation between the results is observed. According to the civil engineering standard, PU materials tested at special conditions (temperature, humidity) should not exhibit more than 3% changes in dimensional linearity [68]. In all cases, the addition of clove filler does not affect the values of Δl, ΔV, and Δm, and the examined PU composite foams are in accordance with commercially acceptable limits [68].

## 4. Conclusions

PU composite foams enhanced with different concentration clove filler (1–5 wt%) were successfully produced. The presented results confirmed that the addition of clove filler influences the rheological behavior, cellular structure, and further mechanical and thermal properties of modified materials. Based on the results, it can be concluded that the addition of 1 and 2 wt% of clove filler leads to PU composite foams with improved compression strength (improvement by ≈18% for sample PU-1), greater flexural strength (increase of ≈11%), and improved impact strength (increase of ≈8%). Moreover, it was proven that clove filler may be used as a natural anti-aging compound for polymeric materials. The incorporation of clove filler in each amount successfully improved the stabilization of PU composite foams. Based on the antibacterial results, it was shown that the addition of clove filler significantly improved the antibacterial properties of PU foams and is suitable for the manufacturing of antimicrobial PU composite foams. Due to these positive and beneficial effects, it can be stated that the use of cloves as a natural filler in PU composite foams will promote a new application path in converting agricultural waste into useful resources for creating a new class of green materials. PU composite foams modified with clove filler may offer solutions based on the main challenges of our times—the use of renewable raw materials, the economy, and the preservation of resources and minimization of the output of waste. This creates an ideal scenario for the emergence of innovative opportunities under rigid quality control procedure, that goes from the raw material to advanced polyurethane composite foams for construction and structural applications.

## Figures and Tables

**Figure 1 materials-13-01108-f001:**
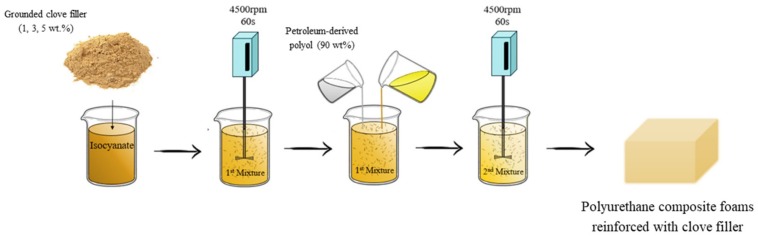
Schematic procedure of the synthesis of polyurethane (PU) composite foams.

**Figure 2 materials-13-01108-f002:**
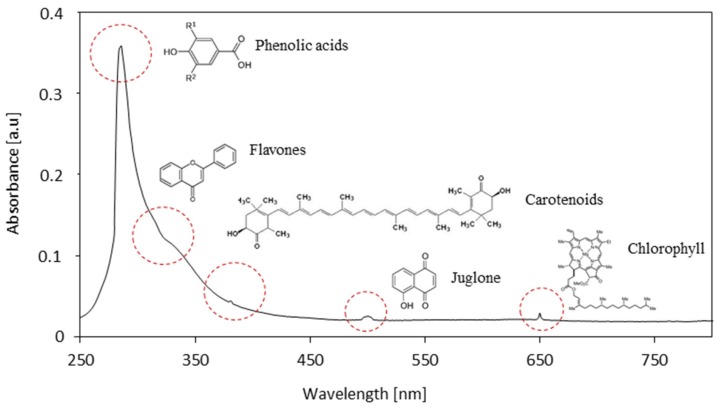
UV-Vis spectra of clove filler.

**Figure 3 materials-13-01108-f003:**
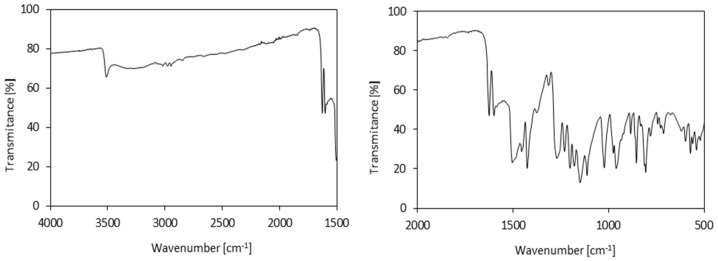
Fourier-transform infrared spectroscopy (FTIR) spectra of clove filler.

**Figure 4 materials-13-01108-f004:**
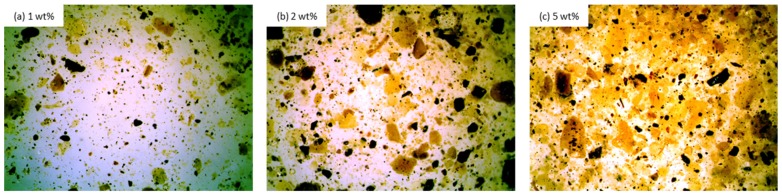
Polyol mixtures containing (**a**) 1 wt%, (**b**) 2 wt%, and (**c**) 5 wt% of clove filler observed at 50× magnification.

**Figure 5 materials-13-01108-f005:**
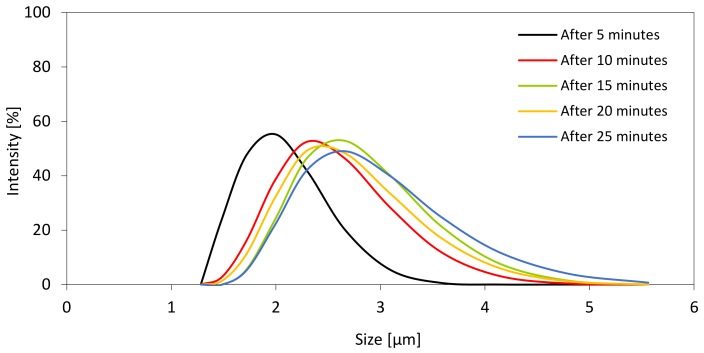
The particle size distribution of the sample measured after 5, 10, 15, 20, and 25 min of ultrasonic mixing.

**Figure 6 materials-13-01108-f006:**
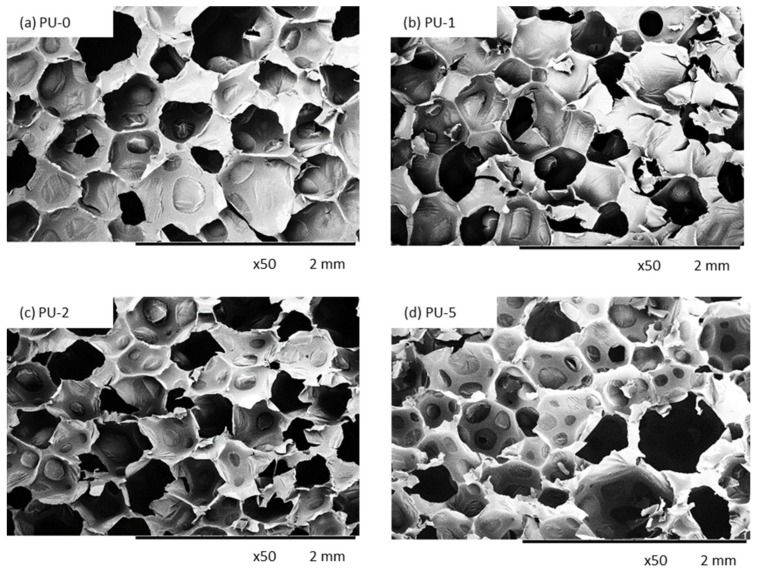
The cellular structure of (**a**) PU-0, (**b**) PU-1, (**c**) PU-2, and (**d**) PU-5 observed at a magnification of 50×.

**Figure 7 materials-13-01108-f007:**
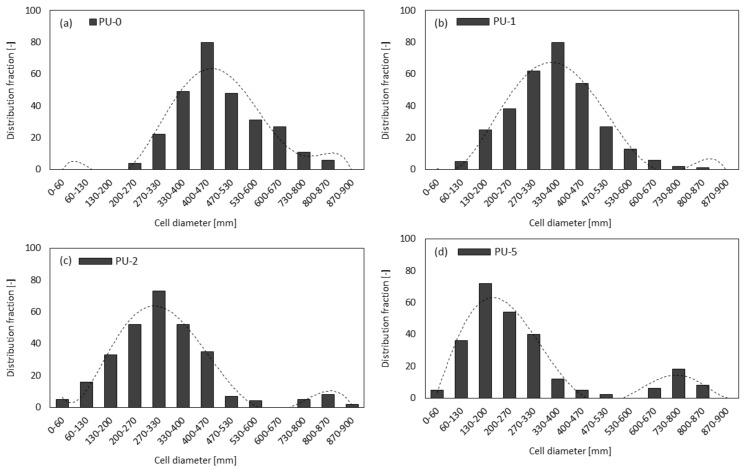
Distribution of cell size of (**a**) PU-0, (**b**) PU-1, (**c**) PU-2, and (**d**) PU-5.

**Figure 8 materials-13-01108-f008:**
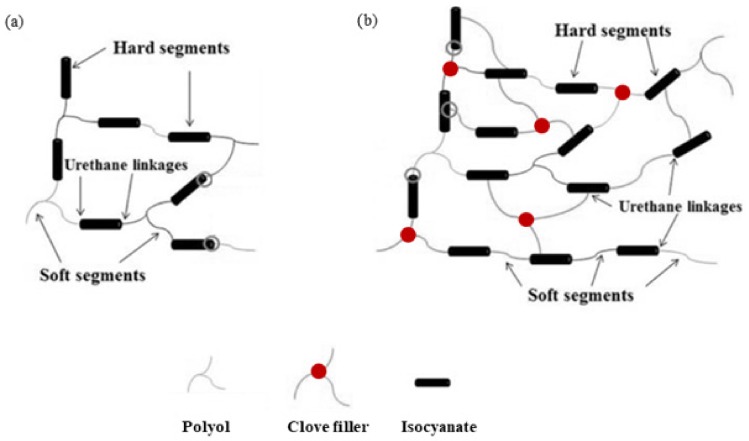
The effect of clove filler on the cross-linking of PU foams: (**a**) without and (**b**) with the addition of clove filler.

**Figure 9 materials-13-01108-f009:**
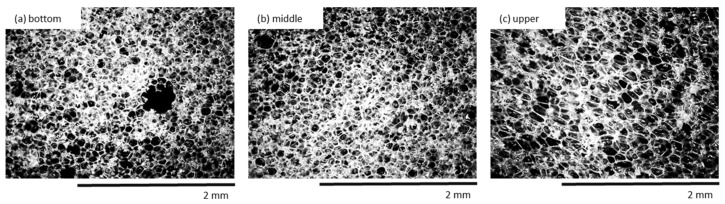
The cellular structure of the (**a**) bottom, (**b**) middle, and (**c**) upper parts of PU-1 observed at a magnification of 50×.

**Figure 10 materials-13-01108-f010:**
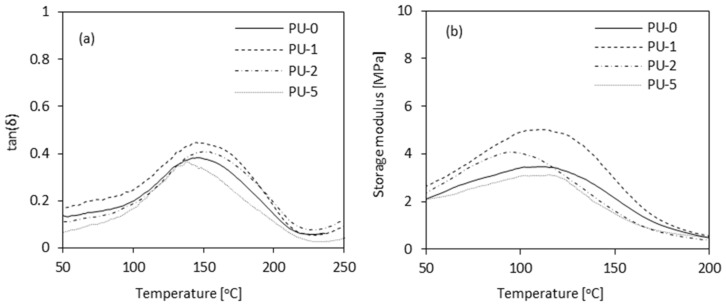
Dynamic mechanical analysis of PU composite foams—(**a**) Tanδ and (**b**) storage modulus as a function of temperature.

**Figure 11 materials-13-01108-f011:**
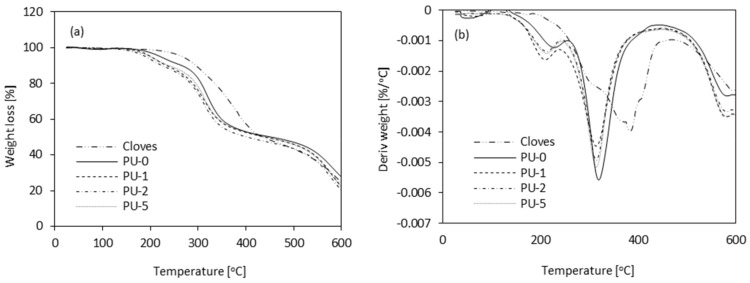
(**a**) TGA (Thermogravimetric Analysis) and (**b**) DTG (Derivative Thermogravimetry) results of PU composite foams.

**Figure 12 materials-13-01108-f012:**
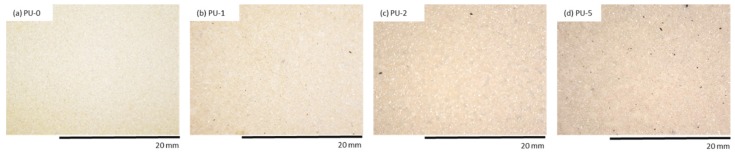
The external surface of (**a**) PU-0, (**b**) PU-1, (**c**) PU-2, (**d**) PU-5 before UV aging.

**Figure 13 materials-13-01108-f013:**
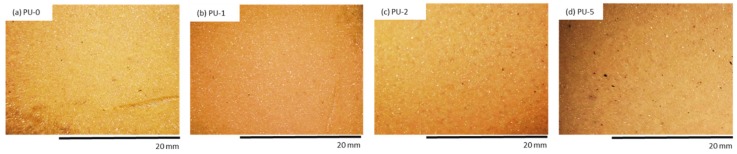
The external surface of (**a**) PU-0, (**b**) PU-1, (**c**) PU-2, (**d**) PU-5 after 7 days of UV aging.

**Figure 14 materials-13-01108-f014:**
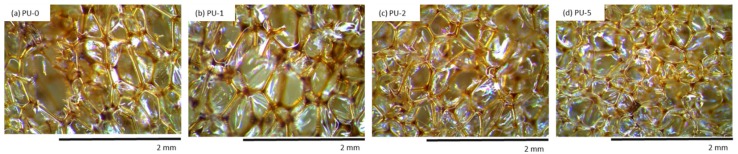
Morphology of (**a**) PU-0, (**b**) PU-1, (**c**) PU-2, (**d**) PU-5 after 7 days of UV aging observed at a magnification of 50×.

**Table 1 materials-13-01108-t001:** Foam formulations.

Foam Symbol	Comments	Mass Content (By Weight)			
		Izopianol 30/10/C	ERGOPLAST^®^ ES	Purocyn B	Clove Filler
PU-0	Reference foam (unfilled)	90	10	160	0
PU-1	Foam reinforced with 1 wt% of clove filler	90	10	160	1
PU-2	Foam reinforced with 2 wt% of clove filler	90	10	160	2
PU-5	Foam reinforced with 5 wt% of clove filler	90	10	160	5

**Table 2 materials-13-01108-t002:** The results of the dynamic viscosity of polyol mixtures measured in the function of shear rates.

Foam Symbol	Dynamic Viscosity *η* (MPa·s)				
	10 RPM	20 RPM	30 RPM	40 RPM	100 RPM
PU-0	650 ± 8	330 ± 8	270 ± 6	160 ± 5	120 ± 6
PU-1	890 ± 8	520 ± 10	360 ± 5	190 ± 5	130 ± 4
PU-2	1390 ± 10	1030 ± 10	840 ± 5	660 ± 8	320 ± 5
PU-5	1850 ± 10	1410 ± 10	1260 ± 6	840 ± 7	540 ± 5

**Table 3 materials-13-01108-t003:** Selected properties of PU composite foams.

Foam Symbol	PU-0	PU-1	PU-2	PU-5
Temperature (°C)	125 ± 2	123 ± 1	122 ± 1	119 ± 1
Cream time (s)	43 ± 4	57 ± 2	58 ± 2	60 ± 2
Expansion time (s)	214 ± 10	228 ± 11	230 ± 8	238 ± 9
Tack-free time (s)	341 ± 14	347 ± 12	351 ± 12	368 ± 10
Cell size (µm)	390 ± 9	380 ± 8	330 ± 6	310 ± 8
Closed-cell content (%)	88 ± 1	90 ± 1	87 ± 1	80 ± 1
Apparent density (kg m^−3^) (bottom)	36 ± 9	36 ± 9	37 ± 9	37 ± 9
Apparent density (kg m^−3^) (middle)	39 ± 1	41 ± 1	42 ± 2	44 ± 2
Apparent density (kg m^−3^) (upper)	38 ± 9	41 ± 9	42 ± 9	44 ± 9

**Table 4 materials-13-01108-t004:** Compressive strength, flexural strength, and elongation of PU composite foams.

Foam Symbol	Compressive Strength σ_10%_ (Parallel) (kPa)	Specific Compressive Strength (Parallel) (kPa kg^−1^ m^−3^)	Compressive Strength σ_10_ (Perpendicular) (kPa)	Specific Compressive Strength (Parallel) (kPa kg^−1^ m^−3^)	Flexural Strength ε_f_ (MPa)	Impact Strength (MPa)
PU-0	250 ± 9	6.4 ± 0.4	144 ± 4	3.7 ± 0.5	0.402 ± 0.008	0.389 ± 0.008
PU-1	296 ± 8	7.2 ± 0.6	176 ± 5	4.2 ± 0.6	0.472 ± 0.010	0.460 ± 0.009
PU-2	283 ± 6	6.7 ± 0.4	148 ± 4	3.9 ± 0.5	0.440 ± 0.009	0.431 ± 0.009
PU-5	210 ± 6	4.7 ± 0.4	130 ± 6	2.9 ± 0.6	0.412 ± 0.009	0.420 ± 0.008

**Table 5 materials-13-01108-t005:** Thermogravimetric analysis (TGA) results of PU composite foams investigated under nitrogen atmosphere.

Foam Symbol	*T_g_* (°C)	*T*_10_ (°C)	*T*_50_ (°C)	*T*_80_ (°C)	Char Residue (%)
PU-0	145 ± 1	265 ± 1	454 ± 1	591 ± 1	28.4 ± 0.2
PU-1	148 ± 1	238 ± 1	440 ± 1	580 ± 1	23.5 ± 0.4
PU-2	152 ± 1	231 ± 1	436 ± 1	572 ± 1	22.2 ± 0.2
PU-5	142 ± 1	226 ± 1	438 ± 1	570 ± 1	20.2 ± 0.2

**Table 6 materials-13-01108-t006:** Microbial properties of PU composite foams against *Staphylococcus aureus* and *Escherichia coli*.

Foam Symbol	*Staphylococcus Aureus*		*Escherichia Coli*	
	Initial Bacterial Suspension (CFU/mL)	Bacterial Suspension Measured after 24 h (CFU/mL)	Initial Bacterial Suspension (CFU/mL)	Bacterial Suspension Measured after 24 h (CFU/mL)
PU-0	74 × 10^6^	74 × 10^6^	74 × 10^6^	74 × 10^6^
PU-1	74 × 10^6^	48 × 10^6^	74 × 10^6^	42 × 10^6^
PU-2	74 × 10^6^	33 × 10^6^	74 × 10^6^	28 × 10^6^
PU-5	74 × 10^6^	18 × 10^6^	74 × 10^6^	16 × 10^6^

**Table 7 materials-13-01108-t007:** Color analysis of PU foams before UV aging.

Foam Symbol	Colorimetric Parameters			
	L*	a*	b*	ΔE*
PU-0	11.7	22.4	−5.1	5.0
PU-1	25.3	72.5	−3.9	18.0
PU-2	49.5	75.6	−3.1	29.6
PU-5	58.5	77.9	0.2	34.1

ΔE*—total color change, L*—degree of lightness, a*—red/green parameter, b*—yellow/blue parameter.

**Table 8 materials-13-01108-t008:** Color analysis of PU foams after UV aging.

Foam Symbol	Colorimetric Parameters			
	L*	a*	b*	ΔE*
PU-0	18.7	24.2	−4.2	22.5
PU-1	28.3	75.4	−3.1	23.2
PU-2	49.5	75.1	−2.5	28.6
PU-5	60.5	79.2	0.3	30.4

ΔE*—total color change, L*—degree of lightness, a*—red/green parameter, b*—yellow/blue parameter.

**Table 9 materials-13-01108-t009:** Changes in linear dimensions (Δl), volume (ΔV), and mass (Δm) before and after conditioning at +70 and −20 °C, contact angle, and water absorption of PU composite foams.

Foam Symbol	Contact Angle (^°^)	Water Absorption (%)	Temperature of +70 °C			Temperature of −20 °C		
			Δl (%)	ΔV (%)	Δm (%)	Δl (%)	ΔV (%)	Δm (%)
PU-0	129 ± 1	11.3 ± 0.8	1.75 ± 0.00	1.40 ± 0.01	1.54 ± 0.01	1.75 ± 0.01	1.65 ± 0.01	1.74 ± 0.01
PU-1	124 ± 2	11.5 ± 0.5	1.58 ± 0.00	1.42 ± 0.01	1.68 ± 0.01	1.74 ± 0.01	1.87 ± 0.01	1.68 ± 0.01
PU-2	121 ± 2	12.8 ± 0.8	1.52 ± 0.00	1.41 ± 0.01	1.90 ± 0.01	1.76 ± 0.01	1.69 ± 0.01	1.88 ± 0.01
PU-5	115 ± 1	14.7 ± 0.6	1.69 ± 0.00	1.90 ± 0.01	1.47 ± 0.01	2.12 ± 0.01	1.79 ± 0.01	1.69 ± 0.01

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
