# Peer review of "Bio-Based Polyurethane Composite Foams with Improved Mechanical, Thermal, and Antibacterial Properties"

_materials, 2020, doi:10.3390/ma13051108_

Round 1
Reviewer 1 Report
In the manuscript the effect of the clove filler on the properties of the PU composite is studied. It is an example of a very detailed and thorough research. Conclusions are supported and controls are performed where needed.
Only minor corrections are suggested:
Section 2.1 Please describe in more details what are the bases for the ratio of isocyanate and polyols chosen in the article. It is not clear for non-specialist.
Through the paper the authors multiple times speculate regarding the nature of the influence of the cloves filler added. Wouldn’t it be useful to measure active OH groups in the cloves filler (mgKOH/g)? I believe it could significantly improve this part of discussion. At least then it would be clear what is the magnitude is the contribution of the reactive groups from the filler.
Part 3.1 The method chosen for the analysis of the cloves filler is rather questionable. However, please add a reference to the literature where particular absorbance bands are stated for the listed compounds. Also please specify where is the isoeugenol place in Fig. 2 and FTIR analysies as it is a major compound of the cloves essential oil/extractives.
Fig. 4, 11, 12, 13 Please add the scale bars.
Fig. 6 Please make the scale bars bigger so they will be easily visible
Reviewer 2 Report
Authors report a new type of polyurethane composite foams consisting of clove-based additives. The work is comprehensive. I suggest the paper be published provided that the following points are addressed.
1. Lines 103-Line 115. Exact recipe must be provided so that the work can be reproducible.
2. Figure 3: Mark the bands that are highlighted in the text in the figure.
3. Figure 4: Scale bar is missing.
4. The size distribution suggests that the fillers have the size of nm. Yet they are visible in Figure 4, which are acquired using optical microscope. This cannot be right because the diffraction limit of the optical microscope limits the size it can detect. Something is seriously wrong with the DLS result.
5. Table 1: there is no uncertainty in this table. The same can be said about the numbers reported in this paper, including the following tables where incomplete uncertainty data are reported. Values without uncertainty are not very meaningful. Please add uncertainties where you can.
6. One major shortcoming of this paper is that the structure of the PU with/without filler, prior to foaming, is not understood. This is important because PU typically has hierarchical structures and structures at the microscopic level affect the macroscopic behaviors. The correlations can be found in a number of recent works, such as (Polymer, 121, Pages 26-37, 2017) and (Polymer International, 64 (11), 1607-1616, 2015). A discussion under this context will be useful.
7. Figure 7: It appears that, with the addition of clove fillers, the cell diameter transits from a unimodal distribution to a bimodal distribution. Please discuss why.
8. Table 5: This table is unreadable. What does PU-O mean in the third and fifth columns?
9. Figure 11-13: include a scale bar. Figure 13 has a significantly different appearance to Fig. 11 and 12. Was it captured using the same camera and the same setup?
10. One frequent issue with this work is that the authors made strong arguments without evidence. An example is the statement between Line 490 to Line 494 “This phenomenon…”. It is completely fine to speculate. But I suggest the authors to go over the paper carefully and tone down the statements when necessary.
Round 2
Reviewer 2 Report
The authors have addressed this reviewer's comment.